# Analysis of Modular Design Applicable in Prosumer Scope. Guideline in the Creation of a New Modular Design Model

**Laura Asión-Suñer *** and **Ignacio López-Forniés**

Department of Design and Manufacturing Engineering, Universidad de Zaragoza, 50018 Zaragoza, Spain; ignlopez@unizar.es

*** Correspondence: lauraasion@gmail.com; Tel.: +34-665-225-201

**Featured Application: Products based on modular design for users who design, manufacture or assemble part of their products, such as prosumers and makers.**

**Abstract:** Modular design is the design based on independent and connectable modules to achieve product variety. It is an ideal tool that could facilitate the inclusion of prosumers in the creative process. However, its evolution has focused on product development and not on end users. The creation of a new modular design model for prosumers could respond to their needs while exploiting the advantages of modularity. The present work analyzes the applicability of modular design in the prosumer scope and defines a guideline for the creation of the new model. To this end, prosumer and modular design methods are collected and analyzed throughout the previously defined design process. The intersection between both terms is analyzed through a series of real cases and design methods that show what the objectives of prosumers are and if the present products and methods meet their needs. These results will establish the current and potential link between modular design and prosumers. Finally, the objectives and stages to develop the design model based on previous results are presented. The paper presents relevant findings such as the lack of methods in the conceptual design phases and a guideline to ensure that the prosumer benefits from modular design.

**Keywords:** modular design; prosumer; design model; end-user; product design; manufacturing

## 1. Introduction

Since the term prosumer was first coined to refer to users who assume the role of a producer and consumer [1], several decades have passed until the technological and social context has propitiated the development of this concept. Factors such as the emergence of digital manufacturing, the growing trend of maker culture or the progressive network of platforms and services have allowed the end user to be included in the creative process through the co-creation, customization and manufacture of their own products [2]. Currently, prosumers use numerous digital tools, both individually and collectively, to be increasingly involved in the design or manufacturing process of a product [3,4]. These actions often respond to needs and wants related to the user, such as self-realization, but also to external causes, such as the development of new technologies [5].

By providing users with effective and intuitive ways of production, the advances in ICT have enabled consumers to create goods in the digital field, thus initiating their transformation to prosumers [6]. From self-consumption to mass customization, we can differentiate several profiles depending on whether consumption (consumer) or production (prosumer) predominates [7]. This work focuses its attention on those users who seek the balance between consumption and production: they create their own physical products by buying part of them and designing/manufacturing the rest. Thanks to this balance, this user profile does not need advanced design and manufacturing knowledge as the product is not developed in its entirety (self-consumption). On the other hand, the level

of production is adequate to facilitate the intervention in the creative process instead of making only a selection of characteristics (mass customization).

Nevertheless, the use of physical tools by prosumers has been practically limited to 3D printing technologies, which have the potential to do the same as digital tools but in the world of physical objects [6]. In this regard, modular design is design based on independent and connectable modules that allow the configuration and variation of a product, making the participation of the end user in the product possible. In this way, it is an ideal physical tool that could facilitate the inclusion of the user in the creative process [8,9]. It can be applied as a design approach tailored to build products with multiple components that enable rapid replacement and customization. In terms of sustainability, modular design can have a positive impact in areas such as environmental management or by prolonging the product life cycle by offering post-market modules [10]. If we focus on the end user, the inclusion of modular design and modules in eco-design principles has proved to be well-received from the consumer's perspective [11,12].

Recent studies show that the key to resolving the conflict between customization and low cost is to implement modular design in physical products and services with the objective of meeting user needs [13,14]. Combined with tools from the prosumer field, such as Open Access, modular design can allow the users to produce only the modules they need [15]. However, its benefits are not restricted only to the pre-consumption phases. Modular design also allows changes to the product through an adaptable interface that enables the inclusion of future updates or accessories [16]. In this way, the prosumer could take advantage of all these characteristics to intervene in the product through the design, manufacture or assembly of his own modules.

Despite the obvious benefits that modular design can provide in the prosumer field, the possibility of focusing its application on this type of user has not yet been studied. Instead, previous research and methods developed for modular design have been geared toward developing and manufacturing the product from the company's point of view [17–19]. This leaves the modular design hidden from the end user, meaning that they can hardly exploit its benefits for their own purposes either in the pre-consumption phase or during its life cycle. When they do, it is only a consequence of modularization in which the user can benefit from some aspect such as customization or component replacement. However, this does not allow a high level of intervention where the user obtains a unique product designed or manufactured by themselves.

The objective of this work is to analyze the current context in both areas and establish a guideline to develop a model for prosumers based on modular design. This is intended to respond to the need to cover the prosumer field from modular design by creating a new design model that allows the end user to design or manufacture their own modules.

To achieve this, the investigation is divided into three sections. First, the process of modular design and the creation of products for prosumers is defined. Moreover, the current methods and tools developed in both areas are analyzed. The second section shows the current and potential application of modular design in the prosumer scope through a series of real cases and design methods developed in recent years. The link between modular design and the prosumer is also defined in this section. Finally, in the third section, guidelines are established for the creation of a new design model based on the previous results.

## 2. Materials and Methods

The present work aims to show the benefits and importance of the creation of a model of modular design focused on prosumers. The investigation begins with the separate analysis of the modular design and prosumer methods developed to date. The compilation of these methods was carried out in both cases through review articles of modular design [8,20] and prosumers [21]. In the case of methods for the prosumer, due to the limited literature and the date of the review, it was decided not to extend the search, concluding that the reference was recent and complete enough. However, in the case of modular design, the selection of

methods was more complex. In this case, the literature is very extensive, going beyond the limits of this study, so it was necessary to filter the methods. To begin the selection, those cited in the review articles were included, excluding those that were repeated. A search was made through more reference articles; then, those that other authors considered essential for the application of the modular design were added to the selection. After that, an absence of methods in the conceptual design phases was detected, so the search was broadened to find modular design methods in these phases. However, the search ended without any results.

To define the applicability of modular design in the prosumer scope, a series of real cases and design methods were exposed. To search for cases, we started from two previous academic works developed in this line of research that collected cases of modular design [22] and prosumer users [3] separately, identifying those that had both in common. Since the resulting number of cases was not representative, the search was extended using keywords such as "customization" or "modular" in technology, industrial design and crowdfunding platforms, such as Kickstarter, Yanko Design, Xataka and Google. Regarding the search for methods, we established as search criteria that the method used modular design to involve the end user in the products, whether to customize, adapt or develop them. We conducted an exhaustive search through different databases (Scopus, Google Scholar, Web of Science, MDPI, Science Direct) with combinations of two groups of keywords: the first on modular design (modular product, modularity, modular design) and the second, in reference to the prosumer (end user, prosumer, lead user, personalization, adaptability). During the search, all results obtained under these criteria were from the last decade because methods with these objectives had not previously been developed, so the search was filtered by years to optimize the results found. Both the real cases and the design methods were analyzed to finally establish the current and potential link between modular design and the prosumer user, taking into account the needs of the prosumer and the advantages of modular design.

## 3. Results

The research results are presented in three sections: the *analysis of current modular design and prosumer methods*, the *use of modular design in the prosumer scope* and *guidelines for the creation of the new design method*. Each one has its own subsections as be introduced at the beginning of each section.

Section 3.1 analyzes separately the prosumer and modular design methods and tools that have been developed to date. This section does not establish a direct relationship between them but serves as a preamble to show the current context in each of the two areas. At the beginning of the section, the design process is defined for both cases.

Section 3.2 covers the current and potential modular design–prosumer relationship. To do this, first a series of real cases where modular design has been focused on prosumer users is presented. Various design methods that meet the same characteristics are also presented, encompassing modular design and prosumer from the same perspective. The section concludes with the definition of the relationship between both terms based on the benefits of modular design and the needs of the prosumer user.

Section 3.3 establishes a series of objectives and stages for the development of future modular design methods focused on prosumers. The objective is to make a series of recommendations that facilitate the creation and implementation of new design models in this area. This section is a preliminary result—a proposal based on the previous results that must be explored and experimented upon.

### 3.1. Analysis of Current Modular Design and Prosumer Methods

This analysis is carried out with a double objective: to verify in which phases of the design process there is currently a lack of modular design and prosumer methods and to identify what current methods can serve as a reference to correctly apply both terms. To achieve this, the design process is defined for each case (modular design and prosumer) and subsequently the methods are situated in these phases to perform an analysis.

### 3.1.1. Definition of the Modular Design and Prosumer Processes

A diagram has been drawn up that defines in detail the process of modular design and product creation for a prosumer (Figure 1). Some models have been taken as a reference that divide a product's life cycle into various phases [23–27]. Specifically, the eight phases defined by Asimow [25,27] have been used because they provide a greater precision within the process than the sub-phases of the general stages (preliminary design, production and consumption) defined by other authors [23,24,26]. Specific objectives of both design processes have been assigned to each phase. This allows us to differentiate between the use of the methods within each general phase based on identified objectives. Thereby, Figure 1 is the basis used to classify the modular design and prosumer methods according to the different phases of the product life cycle in the next sections. This can facilitate the combination of methods from different phases to create a model, but not all sequences are valid. The feasibility of each model must be analyzed since the diagram is only a first step.

| PRODUCT LIFE CYCLE | PRELIMINARY DESIGN | | | | PRODUCTION | | CONSUMPTION | |
|---|---|---|---|---|---|---|---|---|
| | PRIMARY NEEDS | FEASIBILITY STUDY | PRELIMINARY DESIGN | DETAILED DESIGN | PLANNING FOR | | | |
| | | | | | MANUFACTURING | DISTRIBUTION | CONSUMPTION | REMOVAL |
| MODULAR DESIGN | Standarization | Measure modularity; Identify functional groups | | Search for modularity; Functional independence (complex systems); Identify modules; Development of product architectures; Optimize multifunction systems | Optimize manufacturing and assembly; Variety of products; Creation of product platforms; Product family | Transfer by parts (modules) | Perceive brand image; Service optimization | Recyclability analysis; Reuse and recycling |
| PROSUMER | Identify new needs and wants | Choose / vote a new product; Identify and take advantage of trends; Prototype; Study new releases | Generate new ideas; Innovate; Exchange ideas (co-design) | Develop parts of a product (3D, CAD); Customize uniquely; Analyze and Transform (Trial and Error) | Select features; Manufacture at home or workshop (DIY); Learn manufacturing and/or assembly | Share project online | Adapt the product; Update; Hack functions | Reuse and recycling |

**Figure 1.** Phases of the product life cycle and the relationship to modular design and prosumer goals.

As can be seen in the modular design product life cycle (Figure 1), the phases with most design objectives are the detailed design and the manufacturing plan phases. We also found goals in the feasibility study, the consumption plan and the removal plan, while there are no objectives for the primary needs and the preliminary design. On the other hand, in the process of creating products for prosumers, we detected that there is a large variety of objectives and tools in all phases of the design process due to their ability to intervene at all levels. However, its presence is greater in the manufacture and transformation of the product. Specifically, in the pre-design phase, most objectives focus on studying the feasibility of developing a product in order to ensure its success rather than meeting the real needs of the prosumer. This is because these analyses seek to obtain an economic benefit rather than optimize the emotional and social aspect of the product.

3.1.2. Analysis of Modular Design Methods

The relationship between modular design methods and the design phase in which each one is used is analyzed. In addition to locating each method in its phase of use, in Figure 2, we have used three colors to differentiate the essential, interesting and potential methods to achieve the objective set in this research. First, those methods that are considered essential to apply the modular design methodology because they are the basis of the other methods or because of the frequency of their use appear in red. In the case of developing a methodological model, the presence of these methods would ensure the correct application of modular design. Secondly, those methods considered interesting if they are adapted with small modifications to the objective of this research have been highlighted in yellow. Finally, those methods that can potentially be used for both modular design and prosumers at the same time have been highlighted in green.

**Figure 2.** Analysis of current modular design methods according to the phases of the process.

Figure 2 shows a total of 31 modular design methods that have been previously compiled in other review articles on this field [8,20]. The methods presented are not the most relevant at a global level in the field of modular design but are the most present in the cited review articles. As can be seen, an absence of methods is detected in the phases of the identification of primary needs, preliminary design and distribution plan. However, some of the cited methods indirectly affect the distribution phase, although it is not their main objective. The same is not true for the preliminary design phase, where no method has been developed with the aim of being applied in the conceptualization of the product. Instead, most methods operate in the design development phase with an equitable presence between the detailed design and manufacturing planning phases. Finally, we also observe that the presence of methods in the consumption phase is limited, concluding that the end user can hardly take part in the transformation of their own products through modular design.

Four essential methods to apply modular design are detected as they are the basis for the creation of other methods, such as Design Structure Matrix (DSM) or Modular Function Deployment (MFD) [19,28]. This fact is also supported by the number of citations and year of publication of these methods. We observe that two of these aim to identify functional groups (feasibility study), while the other two seek the development of modules to modularize products (detailed design). On the other hand, most of the interesting and potential methods are located in the manufacturing planning phase. Three methods have been identified that could be of interest if they are adapted to other objectives or design phases [29–31]. Two of them are located in the planning for manufacturing phase, and their objectives include developing interchangeable accessories and creating modular products that are user-friendly for the customer. Both methods would be interesting if they evolved to be applied in the consumption phase by the end user. The other method identified is located in the consumer phase and focuses on the brand and sensory esthetics, but it could be of interest if the objective were also functional. Finally, three methods with potential use for prosumers have been detected that can improve their relationship with modular design through product variety, attention to consumer demand and the creation of product families [32–34].

### 3.1.3. Analysis of Methods and Tools for the Prosumer

Figure 3 includes a total of 29 methods and tools used in the creation of products for the prosumer collected in a previous investigation [21]. As in Figure 2, a series of colors has been used to highlight some of the identified methods. In this case, essential methods for the prosumer are highlighted in red, interesting methods in yellow and methods that have a potential use with modular design are highlighted in green. Unlike Figure 2, in Figure 3, there are several methods highlighted in more than one color because they fulfill more than one condition at the same time.

**Figure 3.** Analysis of current methods and tools for the prosumer.

As previously observed in Figure 1, there is a variety of methods throughout all phases of the process. A greater presence is detected in the feasibility study, detailed design and production plan, with these last two phases belonging to the product development stage. The phase with the greatest absence of methods is that of the distribution plan, where only a method based on sharing resources through online platforms and open access is identified—something deeply rooted in the maker philosophy. On the other hand, most of the academic methods are located in the feasibility study, since their objective is to analyze the feasibility of the development of a product through the study of the end users.

Numerous essential methods for creating products for the prosumer have been identified by the high frequency of their use. Although they cannot be considered to be completely essential, they are of great relevance to achieve products aimed at this user category. It is necessary to take this into account if we aim to create a methodological model, since these tools, such as DIY and tutorials, and methods such as redesign are very present in the prosumer methodology. Six methods of interest have also been identified for the future model or design method to be developed, among which 3D printing and methods focused on end consumers stand out. Finally, there are 10 methods and tools that can potentially be used in conjunction with modular design, where five of them are at the same time essential methods. For this reason, we can conclude that the implementation of methods and tools that are both essential and potential are necessary to unite prosumer and modular design in a future method or model. These five methods and tools are new idea/innovation, DIY, adaptation, upgrade and reuse. Specifically, these last three methods are also identified later in *Modular Design–Prosumer Relationship* as common design goals between prosumers and modular design.

### 3.2. Use of Modular Design in Prosumer Scope

These results are presented in three sections: *real cases*, *methods* and *the relationship between modular design and prosumer*. In the first two sections, the information collected is synthesized in tables (Tables 1 and 2) that show the following information: the name of the product or method, author and creation date, brief description, objective pursued and the phase in which the user makes use of modular design. Four possible objectives have been differentiated: *personalization* (P), if the user introduces changes in the product that respond to a desire rather than a need; *adaptation* (A), if the product is modified to meet a need during use; *evolution* (E), if new complementary modules are added or existing ones are updated; or *cost* (C), if the objective is to lower the final price of the product by allowing the user to purchase only the modules they need. Each case or method can meet several objectives at the same time. On the other hand, two phases in which the user can intervene have also been differentiated: *previous phase* to the acquisition of the product (P), which includes stages such as the design, development and manufacture of the product; or *use phase* (U), starting when the user purchases the product until its end of use.

### 3.2.1. Real Cases

Next, Table 1 shows a series of real cases of modular design for prosumers. During the search, it was identified that the cases found could be grouped into four industrial sectors: home automation (H), textile (T), toy (Y) and electronics (E). In this search, all those that only used modularity to optimize manufacturing and product development have been discarded. A total of eight cases have been analyzed—two for each sector. Despite the fact that in toy [35] and electronic [36] sectors, some more cases were found, no more cases have been included because they coincide with those cited in their objectives and intervention phase as they are similar products, so they did not provide more information for the analysis. Finally, the presented cases show examples of products where the end user can intervene in part of the process (design, assembly, evolution, replacement, etc.) through modular design. There are other industrial sectors that have not been included because they do not have a large presence in the market but are in development in the academic

field. This is the case in the automobile sector [32,37] and the medical sector [38,39], which use modular design in order to adapt the product to the end user.

**Table 1.** Modular design real cases for prosumer.

| Product | Author, Date | Description | Objective | Phase |
|---|---|---|---|---|
| (H) Swidwet [40] | Swidget Corp., 2018 (Kanata, ON, Canada) | Smart home device with configurable smart outlets, wall switches and inserts with a compatible connection | A + E | P |
| (H) Freecube [41] | Avatar Controls, 2017 (Irvine, CA, USA) | Command center with interchangeable modules: wireless charger, LED sensor, Bluetooth speaker, power base, etc. | P + E + C | P |
| (T) Ki Ecobe [42] | INNUS KOREA Co., 2017 (Busan, South Korea) | Self-assembled modular shoe with 5 combinable modules (strap, boot, sole, laces and insole), colors and styles | P | P |
| (T) Bloqbag [43] | Bloqbag, 2017 (Jersey City, NJ, USA) | Backpack with pockets of various sizes that are magnetically attached to a base and can be used as individual bags | P + A + E | P + U |
| (Y) Cubelets [44] | Modular Robotics, 2012 (Pittsburgh, PA, USA) | Robot blocks that help teach problem-solving skills with software packages within each hardware modules | A | U |
| (Y) Makeblock Neuron [45] | Distintiva S.Coop., 2017 (Vitoria, Spain) | Modular robotics kit based on 8 modules: battery, Bluetooth, gyroscope, motor, controller, sound and touch | A | U |
| (E) Youmo [46] | Youmo Power, 2016 (Munich, Germany) | Modular multiple charging strip with eleven modules of different energy options (powerline, multi-USB, etc.) | A + E | P + U |
| (E) Fairphone [47] | Fairphone B.V., 2015 (Amsterdam, The Netherlands) | Mobile with replaceable, configurable and upgradeable modules: battery, camera, screen, speaker, case, etc. | E + C | P |

**Table 2.** User-focused modular design methods.

| Method | Author, Date | Description | Objective | Phase |
|---|---|---|---|---|
| Conceptual model for modular service platforms [48] | Løkkegaard, M. et al., 2016 | Design of modular services from a standardized core of modules in order to increase flexibility and adaptability to market changes. | A | P + U |
| Method to optimize assemblability of product earlier [49] | Favi, C.; Germani, M., 2012 | Use of modular design in the initial phase to improve manual assembly with the objective of creating adaptable and customizable products. | P + A | P |
| Methodology for module portfolio planning [13] | Li, H. et al., 2018 | Methodology focused on the product-service system (PSS) that contemplates the construction of a modular structure. | P + C | P + U |
| Personalized product configuration system [50] | Zheng, P. et al., 2017 | Method based on modular and scalable design of the product platform to achieve customization and configuration of the product. | P + A | P |
| Mass customization based on user-experience [16] | Zheng, P. et al., 2017 | Framework that considers three key characteristics of mass customization: user experience, co-creation and product change (modular design). | P + A | P |

### 3.2.2. Methods

Table 2 shows a series of user-focused modular design methods. In total, five methods have been found that met both requirements at the same time. Numerous modular design methods have been developed over the past decades, but it was not until the past decade that prosumer-oriented methods began to be developed. For this reason, the methods presented below correspond to recent research in recent years. Several of them have in common their orientation to personalization and mass customization through modular design.

### 3.2.3. Modular Design–Prosumer Relationship

Through the prior analysis of real cases and modular design methods, the existing and potential relationship between the two areas can be established. Modular design must be treated as a critical tool through which prosumers can achieve specific objectives. We must consider what modular design can offer throughout the life cycle of a product and what the prosumer needs in each phase of this cycle. For this, Figure 4 graphically shows this approach, where the product life cycle has been divided into three main phases [23,24]: *preliminary design phase*, including feasibility study, conceptual design and detailed design; *production phase*, which includes manufacturing and distribution; and the *consumption phase*, which includes the consumption of the product and its disposal. For each phase, we show what modular design can offer, what the prosumer needs and the common points of benefits–needs between the two fields. Within these characteristics, two types of adaptation have been differentiated: *permanent*, which refers to the modifications prior to the purchase made by the user so that the product adapts to the same needs throughout its use life cycle; and *ephemeral*, which corresponds to changes to the product during the consumption phase to adapt to a specific situation, such as changing the lens of a camera.

| PRODUCT LIFE CYCLE | | | |
|---|---|---|---|
| | **PRELIMINARY DESIGN** | **PRODUCTION** | **CONSUMPTION** |
| **D. MODULAR** | · Permanent adaptation<br>· Personalization<br>· Variety of products<br>· Subsequent updates<br>(innovate in separate modules) | · Cost reduction<br>· Ease of assembly<br>· Flexible production<br>· Time reduction | · Ephemeral adaptation<br>· Evolution / accessories<br>· Multifunctionality<br>· Recyclability / Reuse<br>· Maintenance |
| **PROSUMER** | · Personalization<br>· Permanent adaptation<br>· Co-creation (not only<br>choice of modules) | · Ease of assembly<br>· DIY manufacturing | · Ephemeral adaptation<br>· Evolution / accessories<br>· Maintenance<br>· Personalization |
| **BOTH** | · Permanent adaptation<br>· Personalization | · Ease of assembly | · Ephemeral adaptation<br>· Evolution / accessories<br>· Maintenance |

**Figure 4.** Relationship between modular design and the prosumer field.

### 3.3. Guideline in the Creation of the New Design Model

Defining a series of objectives and stages allows a fixed direction to be established and prevents the result from deviating from the main purpose. In addition, these objectives form the basis on which to evaluate whether the final result meets the established goals. This section compiles a series of basic guidelines that the model must comply with so that its application is as effective as possible and takes into account the needs of the prosumers as well as the context in which the modular design is located.

### 3.3.1. Objectives

● **Modular design as a design objective:** It has been detected, both in design methods and in real cases of application, that modular design tends to be used as a tool to solve a problem during product development. In other situations, the product is modularized to make manufacturing more flexible and standardized or to make the production process more economical. Thus, many projects are not conceived as modular design from the beginning, but this design appears as a consequence. It is necessary to consider modular design as a critical Product Design Specification (PDS), conceiving the project as a modular product from the start and, above all, in the conceptualization phase. A correct definition and understanding of the modular

design will make it easier for the user of the model to apply and evaluate it correctly, ensuring that the result is a product based on the modular design process.

- **Conceptual phase of product design (preliminary design):** As shown in Figures 2 and 3, the current presence of prosumer and modular design methods and tools in the preliminary design phase is limited. There are various reasons for this, such as the search for economic benefits for the manufacturer or the frequent intervention of prosumers and makers in the manufacture and consumption of their products. To cover this niche, the model must be conceived for its application in the conceptual phase of product design. This is the most suitable due to the absence of other methods and the need to apply the modular design from the initial stages. In addition, the conceptual phase allows the prosumer to bring more significant and innovative changes and transformations to the final result than during the development, manufacture and use of the product.

- **Relationship with the end user (prosumer):** As has been shown in previous studies [51], prosumers and makers participate in the design and/or manufacture of their products and, subsequently, they continue to evolve them during their use. For this reason, the model should not only take into account the design, manufacturing and assembly phases, but also the later phases of the life cycle. In this regard, modular design gives the products the ability to be upgradable, repairable and adaptable—characteristics directly related to the prosumers and that report important sustainable benefits [10]. Thus, a high usability of the modular design by the end user must be empowered and eased, making it accessible and universal. The objective is that the user can use modular design to intervene throughout the process of creating, producing and using their product, which also implies the design and manufacture of their desired modules.

3.3.2. Stages

- **Conception:** The consolidated presence of modular design methods makes it possible to create a model that evolves current methods towards the creation of products for the prosumer. However, the absence of these in certain phases of the process makes it necessary to create new methods that cover these niches and contribute to the creation of new models focused on these phases. In any case, to reach a new method, it is necessary to first experiment with a model. On the other hand, the homogeneous presence of tools for the prosumer throughout the process means that the combination of prosumer methods (Figure 3) and modular design (Figure 2) in a model may be ideal for covering all aspects of the process. For these reasons, the result can be conceived as a new modular design model aimed at prosumer users whose application will be carried out in the conceptual phases of product design.

- **Development:** Considering that the target user is the prosumer, the model must be able to be used on the one hand by product design professionals and on the other hand by makers and prosumers not specialized in this field. For this reason, the result should not have a high level of complexity—it must be easy to understand and not require advanced knowledge. To achieve this, the result can be presented as an instruction manual, such as a design toolkit [52,53], since it is a format that many users are already familiar with. The manual will be capable of being represented as a tutorial to facilitate its understanding and dissemination, using the graphic resources that are considered necessary for it. As a result, it is proposed to develop a manual in which graphic representations predominate, using, if necessary, a real example to illustrate the model.

- **Application:** According to a previous field study, the majority of prosumers and makers would be willing to use a design model with these characteristics [51]. However, we must ensure that the result is able to meet their expectations in order to consolidate its use. In order to effectively introduce it to the community, the model should pass into the public domain and, therefore, to makers and prosumers. In addition, it must

generate enough interest to facilitate its dissemination among users. On the other hand, to meet the expectations of the users, the result has to be generic enough to be able to be used in various fields. The objective is that the result could be extrapolated so that its use is as wide and adaptable as possible to the needs of each user. This in turn would allow the prosumer to expand their knowledge of product design, making them increasingly specialize until they adopt a more professional profile.

## 4. Discussion

The results show that the methods and tools of modular design are not currently aligned with those of prosumers (Figures 2 and 3). Their exit objectives differ in relevant aspects such as the beneficiary profile (industry vs. consumer) or the level of intervention (massive vs. particular). Figure 4 also shows only some common characteristics between both. Why are there not more points in common? Modular design could include more prosumer-related areas such as co-creation or DIY. However, its current development has not evolved in those directions, although it has the potential to do so due to the benefits it offers. Despite this, it is increasingly common to use modular design as a way to achieve the goals of the prosumer not only in customization, but also in the adaptation and evolution of their own products. It is detected that the boundary between adaptation and personalization is diffuse, since it is difficult to detect the changes that respond to a need (adaptation) from those that respond to a desire (personalization).

Although they are scarce, the current common aspects between modular design and prosumer can form the basis for building a model. It is detected that some connections are influenced by the process or by the design group to which they are directed. While modular design is primarily aimed at professional designers and companies, both individually and collectively, prosumer tools are mostly aimed at individual users. This raises questions such as why there are not more relationships or why there is not a method for this yet. Nevertheless, there is the possibility that there will be more points in common using a new model that poses new relationships. There may still be relationships to be covered between modular design and prosumers due to the lack of connection points that are not currently covered, or others that have not been detected for lack of references.

The design model must take into account the link between modular design and prosumers. It is observed that modular design already covers certain basic needs of the prosumer such as customization, adaptation, evolution or sustainability. However, it still does not allow co-creation by offering only a selection of modules. For this reason, the creative phases are currently not collective. In the case of industry, there may be co-creation by business decisions, although this is not the most common approach. On the contrary, in the case of prosumers, design decisions are mainly individual, and ideas are not shared with others with the same frequency as in the maker field. Could there be an intersection between both that leads to collaboration? Simplifying the design by employing several users or sharing modules could be two possible methods of collaboration. To achieve this, it would be necessary to remove the border between industry and prosumers through a global vision of co-creation.

Considering the topics presented in this work, there are different future research directions that can be continued. The most obvious is the complete development of the proposed model following the guidelines set out. For this, it would also be necessary to define how to introduce it effectively to users by analyzing other similar models [53,54]. On the other hand, the possible limitations of the model should also be analyzed, answering questions such as the proximity to the user/prosumer, applicability in multiple sectors of the industry or its operation with very complex elements, among others. In this regard, the feasibility of applying the method only to the prosumer or to the industry should also be analyzed.

## 5. Conclusions

The analysis of the current context regarding the methods and tools of modular design and prosumers shows a clear lack of methods in the phases of primary needs, preliminary design and distribution plans. While the distribution benefits from some of the methods despite not being the main objective, no method covers the conceptual design phase. This lack is also perceived in the definition of the modular design process, where there are currently no design objectives for these phases. This evidences the need to create a model aimed at these phases, as they are also the phases that allow greater intervention and innovation in the final design of the product. The creation of a new design model in this line would also contribute to the generation of new design objectives in the modular design process that have not been covered until now.

In the real cases analyzed, the most common objectives were the adaptation and evolution of the product, while it was less common to use modular design to reduce costs or customize the product. However, each objective has a certain relationship with each industrial sector analyzed. It has been detected that, in the textile sector, customization is sought-after, and in the home automation and electronic sectors, evolution prevails more; in all sectors, the adaptation to the needs of the user is appreciated. On the other hand, all sectors allow the user to intervene in the phase prior to the acquisition of the product, except toy manufacturers, which only allows this during the useful life cycle. This is because they are ready-made kits that can only be handled once purchased. Some of the cases analyzed also allow the user to intervene in the product during the use phase to respond to ephemeral and variable needs that allow them to better cope with the different situations of use of the product [43,46].

Most of the modular design methods focused on prosumers pursue the customization and adaptation of the products prior to their acquisition. This is because many are focused on applying modularity in mass customization, with the goal that the users may personalize the product, adapting it to their needs and wants before purchase. Only two of the methods contemplate the adaptation of the product or service by the prosumer during the use phase, in which the user can apply modifications, make ephemeral adaptations and evolve or improve the maintenance of the product. Current methods are totally oriented towards customization, but, as has been verified in the case analysis, this is not the main objective of prosumers.

The results show that the new design model must meet three main objectives in order for the prosumer to benefit from modular design. The first is to treat modular design as a design objective (PDS) and not as a tool to solve a problem during the development. The second is to cover the conceptual phases of product design as they are—those in which the user can make a more significant intervention. The third objective is to contemplate the relationship with the prosumer before and during the consumption of the product. On the other hand, to guarantee the effectiveness of the model, it is recommended to present it as an accessible and universal design toolkit. To ensure its consolidation, the model must be applicable in multiple areas and easy to use by non-specialized users.

**Author Contributions:** Conceptualization, L.A.-S. and I.L.-F.; methodology, L.A.-S. and I.L.-F.; investigation, L.A.-S.; writing—original draft preparation, L.A.-S.; writing—review and editing, I.L.-F.; visualization, L.A.-S.; supervision, I.L.-F. All authors have read and agreed to the published version of the manuscript.

**Funding:** This work was supported for publication by the "Conselleria de Innovación, Universidades, Ciencia y Sociedad Digital" of the "Generalitat Valenciana. The authors would like to thank also to the INGEGRAF association.

**Conflicts of Interest:** The authors declare no conflict of interest.

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
