# Peer review of "Analysis of Modular Design Applicable in Prosumer Scope. Guideline in the Creation of a New Modular Design Model"

_applsci, doi:10.3390/app112210620_

Round 1
Reviewer 1 Report
This manuscript presents some good work on analysis of applicability of modular design in prosumer scope. The reviewer finds that there is a significant contribution and innovation to the research industry from the current manuscript. However, the paper needs some modifications before publishing. Please address the below comments.
- Major findings of the study should be described in the abstract
- 8 cases have been used in the study. But not much information is provided for the reader to get the understanding of the cases. Please provide more information
- Although validation was considered in the research, the sample size is very small (8 cases). Please justify.
Author Response
Thank you very much for your interest in our work. We have considered your comments and suggestions to improve the quality of the paper. These modifications have contributed to the improvement of the work, especially in the understanding of the method and the justification of the results.
We have described the major findings of the study in the abstract, specifying the most relevant conclusions of the work.
The information on each case study has been expanded in Table 1. This information has been considered sufficient to introduce each product to the reader, who can obtain more information through the bibliographic references.
Finally, the sample size of the cases analyzed has been clarified and justified. It was not a small selection on a large sample of cases, but a biased search with few results that coincided in their fields of application. The similar cases that shared the same characteristics as those analyzed have been also referenced. Nevertheless, these cases have not been included in the analysis because they did not contribute any new findings to the study.
Reviewer 2 Report
The paper is interesting and well written, however there are some lacks that need to be clarified.
Firstly, the definition of “modular design” is not stated in the paper: On the one hand, I can say that it is obvious for a researcher in this field, but on the other hand I agree with Boinvoisin et al. [8] who stated that “Product modularization fails to get precisely defined because this term describes a large field of practices rather than a defined method, sometimes leading to misunderstandings. A clear and minimal consensus is that ‘modular products are made of modules, building blocks’ (ibid., 296). However, definitions of what a module is tend to diverge. Consequently, there is an unfortunate ambiguity in the use of the terms ‘module’, ‘modularity’ and ‘modularization’.”
So, the authors could provide the readers with a definition according to their investigation and research in this field.
Secondly, it is not clear what are the criteria that guided the “search for methods” described in line 111 and in the followings. In line 115, you said “all valid results” so, in particular, my question is how can you distinguish a valid result from a non-valid one?
You selected 13 review articles [8-20] which are all from 2012 and 2021, except for only one dated back to 2004 (Eppinger). How can you be sure that they are able to include all the relevant methods in that field?
Thirdly, how are the four essential methods of line 196-7 identified? Why (and how) are they selected as “essential” among the others?
Again, in figure 1 and in lines 158, 160 and 172 (caption) is it correct to refer to this process as “design process”? My suggestion is to call it “product development process” or “product life cycle”, because the “real” design process stops after the detailed design, and the next phases can be considered as belong to design only in a sequential process, which is now overcome and obsoleted. In a concurrent approach, all the design phases, such as the manufacturing, assembly, reuse, recycling are anticipated and integrated within the first 4 phases as Design for manufacturing, for assembly, for recycling, etc..
The same idea is expressed by the authors in lines 336-338: “For this 336 reason, the model should not only take into account the design, manufacturing and 337 assembly phases, but also the later phases of the life cycle”.
In general, the paper lacks of content: There are many words spent on methods but they are not really explained as well as the factual description of the case studies are not provided.
So, in some parts, the paper is unnecessarily long, and after many words, it lacks of content.
From the paper, it is not clear how the listed methods work to achieve the goal of product modularization. In each column (of figures 2 and 3), the listed methods are alternative methods in the same product development phase? Again, in each row of the figures, the listed method can work together phase after phase in any combination (sequence) of them?
Again, it would be interesting to have the module design methods classified for product categories (which ones can be used in the toy category? Which ones in the electronics category? etc.).
Some typos or mystakes:dsm
The “incipit” of the abstract is not very fluent: I suggest starting with “Modular design is …”.
line 197: acronyms as DSM, MFD are not introduced;
line 265: electronicS
In line 312 and line 318, “Product Design Specification” is introduced later that its acronym.
line 386: are not instead of aren't
Author Response
Thank you very much for your interest in our work. Your comments and assessment have been of great importance to improve the final version of the paper. We have applied a series of corrections taking into account your evaluation to improve the quality and value of the work.
Following your comments, we have included a definition of modular design according to research both in the text and at the beginning of the abstract.
Regarding the methodological description, the criteria followed in the search for methods have been clarified. The expression "valid results" on line 115 has also been replaced by "results obtained under these criteria" since there are no valid or invalid results, but results that meet or do not meet all the established criteria.
Regarding the review articles [8-20], it has been clarified that “the methods presented are not the most relevant at a global level in the field of modular design, but are the most present in the cited review articles”.
The essential methods of line 196 were chosen for being the basis for the other methods. With a deeper reading of the mentioned methods, we can verify that many of them start from these four methods. This fact is also supported by the number of citations and year of publication of these methods. This explanation has been extended in the paper to clarify why these methods were selected.
Regarding Figure 1, after reading your reasoning, we agree to call it "Product Life Cycle". For this reason, we have replaced it.
Regarding the content of the paper, it has been considered necessary to introduce the methods to justify the conclusions. Since the nature of the article is not to do proofreading work, it is not necessary to explain each method in detail. If in some sections the article may seem long, we have considered it necessary to justify the conclusions, objectives and guidelines set forth below. However, we have taken this observation into account to improve the factual description of the case study.
Figures 2 and 3 analyses modular design and prosumer methods separately, each one with its own goals. Product modularization is just one of these goals, so not all of the listed methods work toward this goal. As for the columns, effectively each one houses different methods that share objectives within the same phase. This makes it possible to combine it with methods from other columns to form a model, but not just any sequence is valid. Although the diagram facilitates this combination, the feasibility of each of them should be analyzed since the diagram is only a first step. This aspect has been clarified before Figure 1, on line 164.
We agree in the interest of classifying modular design methods by product category. That is why we have related it in other previous works [3, 21, 22] and in the section on real cases. However, despite its interest, adding this information in this work could divert the main topic and would not help us to draw new conclusions. This proposal will be taken into account for future work.
Finally, all the cited mistakes about the article have been corrected. Thank you very much for all the questions raised and for your thorough review. It is of great help both for the improvement of the paper and for the general approach of the investigation.